# Apelin Is a Prototype of Novel Drugs for the Treatment of Acute Myocardial Infarction and Adverse Myocardial Remodeling

**DOI:** 10.3390/pharmaceutics15031029

**Published:** 2023-03-22

**Authors:** Sergey V. Popov, Leonid N. Maslov, Alexandr V. Mukhomedzyanov, Boris K. Kurbatov, Alexandr S. Gorbunov, Michail Kilin, Viacheslav N. Azev, Maria S. Khlestkina, Galina Z. Sufianova

**Affiliations:** 1Tomsk National Research Medical Center, Cardiology Research Institute, The Russian Academy of Sciences, Kyevskaya 111A, Tomsk 634012, Russia; 2Branch of Shemyakin-Ovchinnikov Institute of Bioorganic Chemistry, The Russian Academy of Sciences, Pushchino 142290, Russia; 3Department of Pharmacology, Tyumen State Medical University, Tyumen 625023, Russia

**Keywords:** heart, ischemia/reperfusion, apelin, adverse remodeling

## Abstract

In-hospital mortality in patients with ST-segment elevation myocardial infarction (STEMI) is 5–6%. Consequently, it is necessary to develop fundamentally novel drugs capable of reducing mortality in patients with acute myocardial infarction. Apelins could be the prototype for such drugs. Chronic administration of apelins mitigates adverse myocardial remodeling in animals with myocardial infarction or pressure overload. The cardioprotective effect of apelins is accompanied by blockage of the MPT pore, GSK-3β, and the activation of PI3-kinase, Akt, ERK1/2, NO-synthase, superoxide dismutase, glutathione peroxidase, matrix metalloproteinase, the epidermal growth factor receptor, Src kinase, the mitoK_ATP_ channel, guanylyl cyclase, phospholipase C, protein kinase C, the Na^+^/H^+^ exchanger, and the Na^+^/Ca^2+^ exchanger. The cardioprotective effect of apelins is associated with the inhibition of apoptosis and ferroptosis. Apelins stimulate the autophagy of cardiomyocytes. Synthetic apelin analogues are prospective compounds for the development of novel cardioprotective drugs.

## 1. Introduction

The incidence of acute myocardial infarction (AMI) in the USA is 610,000 first-time attacks and 325,000 recurrent attacks [1]. In-hospital mortality in patients with ST-segment elevation myocardial infarction (STEMI) and percutaneous coronary intervention (PCI) is 5–6% [2,3,4]. In recent years, this index has not decreased. The main cause of death in patients with AMI and PCI is cardiogenic shock [5,6]. In-hospital mortality in men with STEMI and cardiogenic shock is 66% [6]. The likelihood of the appearance of cardiogenic shock directly depends on infarct size [7,8]. It is clear that there is an urgent need to develop novel drugs that can limit infarct size and prevent the occurrence of cardiogenic shock. Oxidative stress and the Ca^2+^ overload of cardiomyocytes increase during reperfusion compared to ischemia [9]. Reactive oxygen species (ROS) production is increased in reperfusion and ROS induce mitochondrial permeability transition (MPT) pore opening [9]. This effect promotes the apoptosis of cardiomyocytes [9]. Leukocyte invasion is also involved in reperfusion cardiac injury [9]. In real clinical practice, a patient is admitted with an already existing ischemic myocardial injury in AMI, and treatment should be aimed at preventing reperfusion cardiac injury. Therefore, studies in which apelin was used before reperfusion are of the greatest practical importance. At the same time, studies in which apelin was administered before cardiac ischemia are also of practical importance, since in cardiac surgery using cardioplegic arrest the heart is subjected to ischemia. In this case, administering drugs prior to cardioplegic arrest could be beneficial.

We believe that apelins could become a prototype for creating such drugs for the treatment of stroke and AMI.

## 2. Discovery of the Apelin Receptor and Apelins

In 1993, O’Dowd et al. discovered and cloned a gene which they named *APJ* [10]. *APJ* is located on chromosome 11. This gene encodes G protein-coupled receptor (GPCR) to have transmembrane domains. These domains share their closest identity to the angiotensin receptor (AT1), ranging from 40 to 50% in hydrophobic domains [10]. Other investigators could also clone *APJ* [11]. Angiotensin II does not interact with the APJ receptor. In 1998, an endogenous peptide agonist of the APJ receptor agonist was discovered [12]. This peptide was isolated from bovine stomach extracts and named apelin [12]. A study with a 17-mer mimetic peptide agonist (AMG3054) demonstrated the structural determination of the apelin receptor [13] (Figure 1). 

The precursor to apelin is preproapelin, consisting of 77 amino acid residues, and the apelin sequence is encoded in the C-terminal region of preproapelin [12]. It is believed that preproapelin forms four biologically active peptides: apelin-55, apelin-36, apelin-17, and apelin-13 [14]. Apelin-13 has the highest biological activity (Table 1).

In 2013, a gene encoding a conserved peptide of 32 amino acids was discovered [15]. Investigators named this gene *ELABELA*. Its peptide was named ELABELA or elabela [15]. Pauli et al. then found a peptide toddler, an endogenous agonist to the APJ receptor [16]. Elabela (apela) or toddler is an endogenous agonist of the apelin receptor (APJ receptor) [17]. The *K*i of apela for the apelin receptor was found to be 38.2 nM [17]. Apela mRNA was also found in the rat heart [18]. 

In what cells and tissues are apelins synthesized? Apelin mRNA was found in all organs of the rat [19]. The organs differed significantly in apelin mRNA content. The apelin mRNA levels in lung tissue and mammary glands were approximately 50-fold higher than in renal or myocardial tissue. Immunoreactive apelin-13 was detected in all tissues. The maximum apelin-13 content was found in the lung tissue and mammary glands of rats, apparently because mammary glands contain many adipocytes [19]. The apelin-13 content in the lung tissue was approximately 10-fold higher than in the myocardium. Apelin-13 content in the mammary gland was about 30-fold higher than in the myocardium [19]. Investigators could not detect apelin-13 in rat plasma because the apelin-13 levels in plasma were too low. These data indicate that apelin-13 is not delivered to the organs from the lungs or adipose tissue by blood but is synthesized in situ.

It was found that apelin-12 was synthesized in human and mouse adipocytes [20]. The plasma concentration of apelin-12 in obese humans was about 2-fold higher than in control humans. Apparently, these data served as the basis for calling apelin an adipokine, although apelin-13 is synthesized in large quantities in pulmonary tissue [18]. Pyr^1^-apelin-13 (pyroglutamated 13-amino acid peptide) stimulates brown adipogenesis and the browning of white adipocytes [21]. Pyr^1^-apelin-13 has *K*_i_ = 169 pM for human apelin receptor and *K*_i_ = 64 pM for rat apelin receptor [22]. The non-peptide apelin receptor agonist BMS-986224 has *K*_i_ = 74 pM for human apelin receptor and *K*_i_ = 49 pM for rat apelin receptor [22]. Both apelin receptor agonists stimulated cAMP synthesis in HEK293 cells and activated extracellular signal-regulated kinase 1/2 (ERK1/2) [22].

## 3. Cardiovascular Effects of Apelins

### 3.1. The Expression of Apelins and Their Receptor in Rats and Humans

Apelin-12 was detected in human left ventricular tissue [23]. Apelin-12 is localized in the endothelial and smooth muscle cells of coronary arteries and also in cardiomyocytes [23]. The immunohistochemical method was used for detection of apelin and the apelin receptor in the rat heart [24]. Western blot was also used to measure the apelin levels. High apelin levels were found especially in endothelial cells and perivascular mast cells. Apelin was not detected in cardiomyocytes or fibroblasts. The apelin receptor was found in endothelial cells, cardiomyocytes, and vascular smooth muscle cells but not in fibroblasts and mast cells [24]. The endogenous agonist of the apelin receptor apela mRNA was detected in the rat heart [18]. It was demonstrated that the apelin receptor is expressed in the atrial tissue and heart ventricles of rats [25] and in the human heart [26]. The apelin receptor was identified in the epicardial coronary arteries of rats [27]. 

Apelin is synthesized by cardiac endothelial and mast cells. The apelin receptor is expressed by endothelial cells, smooth muscle cells, and cardiomyocytes. 

### 3.2. The Effects of Apelins on Blood Pressure and Heart Rate

It was reported that apelin-13 can decrease blood pressure (BP) and increase heart rate (HR) [28,29,30,31,32,33] (Figure 2). 

Apelin-13 was injected intravenously at a dose of 6 µg/kg in anesthetized rats. This peptide triggered a transient reduction in BP without significant alteration of HR [28]. Pyr^1^-apelin-13 (10 nmol/kg = 15 µg/kg) increased HR in anesthetized rats by 11 beats/min and had no effect on HR in rats in an awake state [29]. Pyr^1^-apelin-13 at a dose of 65 nmol/kg (98 µg/kg) increased HR by 14 beats/min in rats in an awake state. Pyr^1^-apelin-13 at a dose of 10 nmol/kg had no effect on BP and reduced BP at a dose of 60 nmol/kg in anesthetized rats. Both effects were transient [29]. Apelin-12 at a dose of 0.07 µmol/kg (0.1 mg/kg) intravenously decreased BP by 20% at a dose of 0.35 µmol/kg (0.5 mg/kg) reduced BP by 40% [34]. This effect was transient. Intravenous infusion of Pyr^1^-apelin-13 induced forearm vasodilation in patients with heart failure and control subjects [35]. Acetylcholine- but not apelin-13-induced vasodilation was reduced in patients with heart failure. Intracoronary injection of apelin-36 increased the maximum rate of rise in left ventricular pressure and coronary blood flow [35]. Intravenous infusion of Pyr1-apelin-13 increased the cardiac index, lowered BP, and reduced peripheral vascular resistance in control subjects and patients with heart failure. Apelin increased HR only in healthy subjects [35].

The hypotensive effect of apelins is apparently the result of their direct effect on arteries. Pyr^1^-apelin-13, apelin-13, and apelin-36 induced a concentration-dependent vasodilatation of isolated endothelium-intact human mammary arteries preconstricted with endothelin-1 [36]. EC_50_ was 0.6 to 1.6 nM. Arterial rings (120–150 µm; 1.2 mm in length) were prepared from coronary arteries of rats [27]. Rings were precontracted with 5-hydroxytryptamine. Apelin-13 induced vasodilation of the coronary arteries at the final concentration of 10 nmol/L. The apelin receptor antagonist F13A and the NOS inhibitor nitro-L-arginine (NLA) abolished apelin-induced vasodilation. 

Thus, NOS plays a key role in apelin-induced vasodilation. 

### 3.3. The Effect of Apelins on the Contractile Function of the Heart

In a study with an isolated rat heart, infusion of apelin-16 at a final concentration of 0.01 to 10 nmol/L induced a dose-dependent positive inotropic effect (EC_50_: 33 pmol/L) [37] (Figure 2). The positive inotropic effect of apelin-16 peaked at 1 nmol/L. U-73122, a phospholipase C (PLC) inhibitor, completely abolished any positive inotropic effect of apelin-16. Staurosporine and GF-109203X, protein kinase C (PKC) inhibitors, mitigated but did not eliminate the apelin-16-induced inotropic effect. The Na^+^-H^+^ exchange (NHE) isoform-1 inhibitor zoniporide and Na^+^-Ca^2+^ exchange (NCX) inhibitor KB-R7943 mitigated but did not abolish an inotropic effect of apelin-16. Apelin-16 had no effect on L-type Ca^2+^ current or voltage-activated K^+^ currents in isolated adult rat ventricular myocytes [37]. Consequently, a key role in the inotropic effect of apelin-16 belongs to PLC. PKC, NHE, and NCX also are involved in the positive inotropic effect of apelin. In a study with isolated electrically stimulated rat cardiomyocytes, it was found that apelin-16 (1 nmol/L) increases [Ca^2+^]_i_ transients and fractional shortening (FS) [38]. Apelin-16 increased the total cytosolic Ca^2+^ concentration and reduced sarcoplasmic reticulum (SR) Ca^2+^ content. Apelin-16 increased SR Ca^2+^-ATPase (SERCA) activity, and this effect was completely abolished by chelerythrine. It was concluded that PKC and SERCA are involved in the positive inotropic effect of apelin-16 [38]. 

Intravenous administration of apelin-13 (40 and 60 μg/kg) induced transient (5–6 min) positive inotropic and hypotensive effects in renovascular hypertensive (2K1C) rats [39]. Pretreatment with the κ-opioid receptor (κ-OR) antagonist nor-binaltorphimine, the G_i/o_ proteins’ inhibitor pertussis toxin, and chelerythrine abolished a positive inotropic effect of apelin-13 (60 μg/kg). Investigators concluded that there is heterodimerization between the κ-OR and apelin receptor [39]. It was found that intravenous injection of Pyr^1^-apelin-13 (10 µg/kg, intravenously) increased cardiac output and stroke volume in anesthetized rats by 15% and 12%, respectively [22]. An increase in dose up to 100 µg/kg did not increase the positive inotropic effect of Pyr^1^-apelin-13.

These data indicated that apelins exhibit a positive inotropic effect both in vivo and in vitro at the level of the isolated heart or isolated cardiomyocytes. This effect is mediated via PLC, PKC, G_i/o_ proteins, κ-OR, ERK1/2, and SERCA. The role of NHE and NCX in the inotropic effect of apelin is minimal. Investigators did not use the apelin receptor antagonist; therefore, it is unclear whether this effect is mediated via activation of the apelin receptor in cardiomyocytes.

### 3.4. The Cardioprotective Effect of Apelins in Ischemia and Reperfusion of the Heart 

The isolated mouse heart was subjected to global ischemia (30 min) and reperfusion (35 min) [40]. The heart was perfused with a solution containing apelin-13 (1 µmol) during reperfusion (35 min). Apelin-13 reduced infarct size by about 40%. Mice underwent coronary artery occlusion (CAO, 30 min) and reperfusion (120 min). Injection of apelin-13 (1 mg/kg) in reperfusion contributed to infarct size reduction by about 40%. An increase in a dose of apelin did not increase the infarct-reducing effect of apelin-13. Apelin-36 exhibited a weaker infarct-sparing effect than apelin-13 [40]. Thus, apelins limited infarct size in reperfusion through a direct effect on the heart. The isolated perfused rat heart was subjected to CAO (35 min) and reperfusion (30 min) [41]. The heart was perfused a solution with Pyr^1^-apelin-13 (10 nmol/L) prior to ischemia or after ischemia. Apelin reduced infarct size in reperfusion by about 15% and had no effect if it was used prior to ischemia. Apelin did not improve the contractile function of the heart in reperfusion [41]. The isolated rat heart was subjected to global ischemia (40 min)/reperfusion (30 min) [42]. The heart was perfused with Krebs–Henseleit buffer containing apelin-13 (30 pmol/L). Apelin-13 improved the recovery of the contractile function of the heart, reduced lactate dehydrogenase (LDH) release in the coronary effluent, and decreased malondialdehyde (MDA) content in myocardial tissue. Surprisingly, apelin-13 at such a low concentration (30 pmol/L) showed a cardioprotective effect. This concentration was lower than the *K*_i_ (64 pmol/L) for apelin [22]. Apparently, this result is a statistical error associated with the use of the small group of animals (*n* = 6). 

The isolated perfused rat heart was then subjected to global ischemia (35 min) and reperfusion (30 min) [43]. The heart was perfused with a solution containing apelin-12 (35, 70, 140, 280, and 560 μmol/L) prior to ischemia or at the onset of reperfusion. Apelin-12 improved the recovery of contractile function after ischemia. The effect reached its maximum at a final concentration of apelin-12 of 140 μmol/L (*n* = 8) [43]. Apelin-12 had a more pronounced inotropic effect if it was infused prior to ischemia. Infusion of apelin-12 (140 μmol/L) prior to ischemia promoted an increase in ATP levels by 58%, induced a reduction in lactate content in myocardial tissue by 20%, and reduced LDH release in reperfusion [43]. Investigators hypothesized that the cardioprotective effect of apelin-12 could be due to improved energy metabolism. Rats were subjected to CAO (40 min) and reperfusion (60 min) [34]. Intravenous administration of apelin-12 at a dose of 0.07 µmol/kg (0.1 mg/kg) reduced infarct size by 21%, at a dose of 0.35 µmol/kg (0.5 mg/kg) by 34% in reperfusion [34]. It was found that apelin-12 at a dose of 0.35 µmol/kg (0.5 mg/kg, intravenously) reduced infarct size by 40% in rats with reperfusion of the heart after CAO (40 min) [44]. Apelin-12 reduced infarct size by 33% at a dose of 0.7 µmol/kg (1 mg/kg). The isolated rat heart was subjected to global ischemia (30 min) and reperfusion (120 min) [45]. The heart was perfused with a solution containing apelin-13 (0.5 µmol/L) prior to ischemia (20 min) or in reperfusion (20 min). Pretreatment with apelin-13 before ischemia did not reduce infarct size. Infusion of apelin-13 reduced infarct size by about 60% in reperfusion [45]. Apelin-13 (0.1. 1, and 10 µg/kg) was injected intravenously after CAO (30 min) 15 min prior to reperfusion (2 h) in rats [46]. Apelin-13 at a dose of 0.1 µg/kg reduced infarct size by about 30%; this peptide at a dose of 1 µg/kg decreased infarct size by approximately 40%; and at a dose of 10 µg/kg, it reduced infarct size by about 30% [46]. Apelin-13 (1 µg/kg) reduced a number of TUNEL-positive (apoptotic) cells in myocardial tissue, decreased caspase-3 activity, and reduced the levels of phosphorylated c-Jun N-terminal kinases (p-JNK) and cleaved-caspase-12 [46]. Consequently, apelin-13 prevents necrosis and apoptosis in the reperfusion of the heart. The use of too high of a dose of apelin-13 may lead to the disappearance of the infarct-reducing effect. 

A comparative analysis of the cardioprotective properties of apelin-12 and its analogue N^α^-MeArg-Pro-Arg-Leu-Ser-His-Lys-Gly-Pro-Nle-ProPhe-OH (AI) was performed in a study with a rat heart subjected to ischemia/reperfusion (I/R) both in vivo and in vitro in reperfusion [47]. It was found that both peptides (0.35 µmol/kg) exhibited an identical infarct-reducing effect in vivo. Both peptides improved the recovery of contractile function of the heart in reperfusion in vitro [47]. Myocardial infarct could affect apelin and apelin receptor expression in the heart. Permanent CAO induced a reduction in the apelin and apelin receptor levels in myocardial tissue 7 days after coronary artery occlusion in mice [48]. Apelin knockout promoted an increase in infarct size by about 36%, reduced survival of mice with permanent CAO, and aggravated apoptosis in the heart after CAO. A disturbance of apelin expression contributed to a decrease in the p-Akt and p-ERK1/2 levels in myocardial tissue after myocardial infarction (MI). Apelin knockout enhanced neutrophil and macrophage invasion in the mouse heart after CAO, increased TNF-α, IL-1β, and IL-6 content in myocardial tissue and aggravated contractile dysfunction after MI [47]. A disturbance of apelin expression disrupts the recovery of contractile function of the isolated mouse heart after ischemia (30 min) and reperfusion (40 min). Pyr^1^-apelin-13 analogue II improved the recovery of contractile function of the isolated heart. Apelin knockout reduced hypoxia-inducible factor-1α (HIF-1α) and VEGF and suppressed myocardial angiogenesis after MI [48]. Consequently, apelins increase cardiac tolerance to I/R and participate in myocardial angiogenesis after MI.

Isolated rat hearts were subjected to global ischemia (30 min) and reperfusion (30 min) [49]. Hearts were perfused with a solution containing apelin-13 (1 nmol/L^–1^ µmol/L) before and after ischemia. Apelin (0.1–1 µmol/L) reduced LDH release, improved recovery of the contractile function of the heart in reperfusion, decreased MDA, nitrotyrosine, and lactate content in myocardial tissue, and increased the reduced glutathione level in the heart. Apelin restored sarcoplasmic reticulum Ca^2+^-ATPase (SERCA) activity and ^3^H-ryanodine binding [49]. Consequently, the improvement in the contractile function of the heart could be a result of increased SERCA activity and ryanodine receptor density in cardiomyocytes.

Mice underwent CAO (45 min) and reperfusion (24 h) [50]. Apelin-13 was injected intravenously (0.1 µg/kg) 5 min after the onset of reperfusion. Its peptide reduced infarct size by approximately 30% [50]. The isolated rat heart subjected to global ischemia (40 min) and reperfusion (30 min) was perfused with St. Thomas’ Hospital cardioplegic solution No. 2 (STH2) [51]. Apelin-12 and its analogues (AI or AII) were added to cardioplegic solution before ischemia or after ischemia at the final concentrations of 70, 140, and 280 µmol/L. All peptides improved recovery of the contractile function of the heart in reperfusion and reduced LDH release. The inotropic effect reached its maximum at 140 µmol/L. A further increase in apelin concentration led to a decrease in this effect [51]. The isolated rat heart was subjected to hypoxic perfusion (4 mL/min, 5 min) + global ischemia (35 min) and hypoxic reperfusion (4 mL/min, 5 min) + reperfusion (25 min) [52]. Both apelin-12 (140 µmol/L) and apelin-13 (140 µmol/L) induced a completely identical recovery of contractile function and ATP content in the myocardial tissue, reducing the lactate levels in the heart. Both peptides reduced DMPO-OH adduct concentration in coronary effluent and suppressed LDH release. Hypoxia (16 h) induced the apoptosis of H9C2 cells and triggered superoxide (O_2_^•^) generation in mitochondria. Apelins decreased a number of TUNEL-positive cells and O_2_^•^ production [52]. 

Consequently, apelins increased cardiac tolerance to I/R and prevented the oxidative stress, necrosis, and apoptosis of cardiomyocytes. Apelin-13 (0.5 µmol/L) reduced infarct size by about 50% and improved the recovery of contractile function of the isolated rat heart in reperfusion [53] (Figure 2). Apelin-13 (10 nmol/kg/day = 15 µg/kg/day, intraperitoneally) was administered to rats for 5 days; then, rat hearts were isolated and subjected to CAO (30 min) and reperfusion (55 min) [54]. Pretreatment with apelin-13 reduced infarct size by 30% (Table 2). 

### 3.5. Apelins Prevent Adverse Myocardial Remodeling

Apelin-13 (1 mg/kg/day) was injected intraperitoneally for 3 days before permanent CAO and for 14 days after CAO [55]. Apelin-13 reduced infarct size and decreased a number of TUNEL-positive cells 24 h after CAO by about 40%. Apelin-13 promoted an increase in the p-Akt, p-eNOS, and VEGF levels in myocardial tissue 24 h after CAO. Apelin triggered the homing of CD133^+^/c-Kit^+^/Sca1^+^ vascular progenitor cells in the infarcted heart. Apelin-13 increased myocardial capillary and arteriole density 14 days after MI. Chronic administration of apelin-13 prevented cardiac hypertrophy and improved cardiac contractile function 14 days after CAO [55]. Apelin-13 (1–100 nM) inhibited collagen synthesis by mouse cardiac fibroblasts [56]. The molecular mechanism of this effect is unclear because cardiac fibroblasts do not express the apelin receptor [24]. Apelin-13 stimulated the proliferation of H9c2 cardiomyoblasts [57]. The maximum effect was observed at a concentration of 200 nmol/L. The effect disappeared at a concentration of 800 nmol/L [57]. These data are credible because the apelin receptors are expressed on cardiomyocytes [24].

Apelin-13 was injected intraperitoneally at a dose of 20 nmol/kg/day to rats with permanent CAO for 28 days [58]. Chronic administration of apelin-13 improved the contractile function of the heart 28 days after MI and reduced infarct size. Investigators wrote that they used Pfeffer’s method for the evaluation of infarct size. However, Pfeifer et al. wrote that they measured fibrous infarct 14 days after CAO [59]. Therefore, it should be more correctly written that apelin decreased scar size. Apelin-13 at a final concentration of 1 µmol/L increased the proliferation of cardiac microvascular endothelial cells.

Rats underwent CAO (30 min) and reperfusion (14 days) [60]. Pyr^1^-Apelin-13 was injected intraperitoneally at a dose of 10 nmol/kg/day (15 µg/kg/day) for 5 days beginning 24 h after CAO. Pyr^1^-apelin-13 improved the contractile function of the heart and reduced the fibrosis area by about 60%. Apelin increases the VEGF mRNA, VEGF receptor-2 (Kdr) mRNA, and angiopoietin-1 (Ang-1) mRNA, eNOS, and mRNA levels in myocardial tissue. In addition, chronic administration of Pyr^1^-Apelin-13 prevents apoptosis of cardiomyocytes [61]. Investigators concluded that Pyr^1^-apelin-13 exhibits angiogenic and anti-fibrotic effects via the formation of new blood vessels and enhancement of the expression of VEGFA, Kdr, Ang-1, and eNOS in the infarcted myocardium [60]. 

Rats underwent permanent CAO [62]. Adverse cardiac remodeling was developed 28 days after CAO. Apelin-13 (10 nmol/kg/day, intraperitoneally) was administered for 28 days. Myocardial infarction resulted in an increase in apelin and apelin receptor expression by about 2-fold. Chronic administration of apelin-13 prevented the development of contractile dysfunction and cardiac fibrosis. In a study with isolated cardiac fibroblasts, apelin-13 suppressed angiotensin II-induced collagen synthesis by these cells. Apelin-13 reduced the PI3-kinase and p-Akt levels in cardiac fibroblasts stimulated by angiotensin II. Myocardial infarction (28 days) simulated NADPH oxidase activity and increased superoxide production in the heart. Chronic administration of apelin-13 reversed these alterations. Investigators concluded that apelin-13 prevents adverse post-infarction remodeling of the heart through the inhibition of oxidative stress and suppression of the PI3-kinase/Akt pathway in cardiac fibroblasts [62]. 

The endogenous apelin receptor agonist apela (1 mg/kg/day) was continuously administered to mice with permanent CAO for 2 weeks by an osmotic minipump [63]. Apela reduced serum N-terminal brain natriuretic peptide (NT-proBNP) concentration and increased the left ventricular ejection fraction (LVEF) 2 and 4 weeks after MI. Apela reduced infarct size and interstitial fibrosis 4 weeks after MI. The accuracy of infarct size measurement 4 weeks after permanent CAO is questionable because 4 weeks after MI, a scar is formed in the infarcted myocardium [64]. Apela increased the expression of neovascular endothelial cell marker CD31 by about 20%. Apela reduced a number of TUNEL-positive and myeloperoxidase-positive cells [63].

Apelin can prevent adverse myocardial remodeling not only after permanent CAO but also after pressure overload. Apelin knockout and wild-type mice were subjected to pressure overload by aortic banding [65]. Aortic banding promoted an increase in the apelin content in myocardial tissue in wild-type mice. A loss of apelin had no significant effect on overload-induced hypertension and cardiac hypertrophy but impaired cardiac contractility [65]. Chronic administration of the apelin receptor agonist BMS-986224 to rats with renal hypertensive-induced cardiac hypertrophy and heart failure improved contractile function [22]. Chronic infusion of angiotensin II (1.5 mg/kg/day) with an osmotic minipump for 14 days led to the development of hypertension, cardiac hypertrophy, and cardiac fibrosis in mice [66]. Chronic administration of elabela-32 (1 mg/kg/day) abolished these negative effects of angiotensin II. The inhibitor of ferroptosis, ferrostatin-1 (1 mg/kg/day), resulted in the same positive effects [66]. Both elabela and ferrostation-1 abolished angiotensin II-induced increases in the MDA levels in myocardial tissue. It was concluded that the cardioprotective effect of elabela is associated with the inhibition of ferroptosis [66].

Thus, activation of the apelin receptor prevents the development of adverse remodeling of the heart induced by MI or pressure overload (Figure 2).

### 3.6. The Effects of Apelins on Regulated Cell Death

There are five main regulated forms of cell death: apoptosis, autophagy, pyroptosis, necroptosis, and ferroptosis. It has been found that apelins exhibit anti-apoptotic properties in I/R of the heart [42,46,52,61,67,68]. Apelin-13 (30 pmol/L) abolished the hypoxia/reoxygenation-induced apoptosis of isolated neonatal cardiomyocytes [42]. Rats underwent CAO (30 min) and reperfusion (4 h) [68]. Elabela (700 µg/kg) was injected intravenously 5 min before reperfusion. Elabela reduced collagen expression and decreased a number of TUNEL-positive cells in myocardial tissue. Elabela increased the ATP and glutathione (GSH) levels in the heart and reduced ROS production and MDA content in myocardial tissue [68]. 

These data indicate that apelins have an anti-apoptotic effect in I/R of the heart. This effect could be involved in the cardioprotective effects of apelins. 

Specific markers of ferroptosis do not exist [69]. This form of cell death is accompanied by a rupture of the cell membrane. An increase in MDA and 4-hydroxynonenal levels is an indicator of ferroptosis [69]. It was demonstrated that apelins reduced MDA levels in myocardial tissue and suppressed oxidative stress in I/R [42,49,68]. The cardioprotective effect of elabela is associated with the inhibition of ferroptosis in mice with pressure overload remodeling of the heart [66]. Consequently, apelins could inhibit ferroptosis in the heart. 

However, convincing evidence is required to show that this inhibition is directly related to the cardioprotective effects of apelins.

Autophagy is a form of cell death without a rupture of the cell membrane [70]. There is evidence that the activation of autophagy increased cardiac tolerance to I/R [71,72,73]. Apelin-13 triggered hypertrophy in H9c2 [74]. This effect was associated with the activation of autophagy. Intramyocardial injection of adenovirus-apelin stimulated autophagy in mice with streptozotocin-induced diabetes [75]. Apelin-13 activated the autophagy of HL-1 cells pretreated with angiotensin II through the activation of AMPK and inhibition of mTOR [76].

Thus, there is evidence that the cardioprotective effects of apelins are associated with the inhibition of apoptosis, ferroptosis, and stimulation of autophagy (Figure 2). The roles of necroptosis and pyroptosis in the apelin-induced tolerance of the heart remain unclear.

## 4. The Signaling Mechanism of the Cardioprotective Effect of Apelins

In a study with transfected COS cells expressing G_i_ proteins, it was demonstrated that apelin-13 and apelin-36 stimulate G_i1_ and G_i2_ proteins, inhibit adenylyl cyclase, and cause the phosphorylation of ERK or Akt-kinase [77]. Apelin-13 activated G_q_ proteins and ERK in HEK293 cells [78]. There is evidence that apelin can activate G_s_, G_q11_, and G_12/13_ proteins [79,80]. It was reported that the apelin receptor interacts with G_i/o_ protein in bullfrogs [81]. The interaction of the apelin receptor with G_i_ protein was confirmed by Deng et al. [82]. The apelin receptor agonists Pyr^1^-apelin-13 and BMS-986224 stimulate G_i/o_ proteins and G_12/13_ proteins in HEK293 cells [22]. 

Kinases play an important role in the regulation of cardiac tolerance to I/R. Apelin-13 limited infarct size in the reperfusion of the isolated mouse heart [40]. The PI3-kinase inhibitor LY294002 and the ERK1/2 inhibitor U0126 abolished an infarct-reducing effect of apelin-13. Apelin-13 stimulated the phosphorylation of PI3-kinase and Akt-kinase, had no effect on the phosphorylation of eNOS and reduced AMPK phosphorylation [40]. The involvement of ERK1/2 and Akt in the infarct-limiting effect of apelin-13 was confirmed by Smith et al. [83]. These data demonstrated that the cardioprotective effect of apelin-13 is mediated via the activation of PI3-kinase, Akt, and ERK1/2. It was reported that the PI3-kinase inhibitor wortmannin and the mammalian target of rapamycin (mTOR) kinase inhibitor rapamycin did not abolish an infarct-sparing effect of Pyr^1^-apelin-13 in reperfusion of the isolated rat heart [41]. These data indicate that PI3-kinase and mTOR are not involved in the cardioprotective effect of apelin. 

It was demonstrated that NO-synthase is involved in the cardioprotective effect of apelin-12 in the reperfusion of the heart both in vivo and in vitro [43,44]. Infusion of apelin-13 (0.5 µmol/L) protects isolated rat hearts against reperfusion injury [45]. The NOS inhibitor L-NNA abolished this protective effect [45]. It was found that the PI3-kinase inhibitor LY294002, the NOS inhibitor L-NAME, and the ERK1/2 inhibitor PD98059 completely abolished the infarct-reducing effect of apelin-13 at the reperfusion of the rat heart in vivo [46]. The AMPK inhibitor compound C did not alter the cardioprotective effect of apelin-13 [46]. However, there is evidence that apelin-13 increased cardiac tolerance to the cardiotoxic effect of bupivacaine through the activation of AMPK [84]. 

Apelin-12 reduced the MDA levels in myocardial tissue and increased superoxide dismutase (SOD), glutathione peroxidase and catalase activity in the heart in reperfusion both in vivo and in vitro [47,85]. Apelin-12 and its analogue AI reduced 5,5-dimethyl-1-pyrroline-N-oxide (DMPO) adduct levels in coronary effluent in reperfusion [47,85]. These data indicate that apelin-12 and AI reduce free radical generation and prevent oxidative stress in the rat heart in reperfusion. Wang et al. also found “an antioxidant effect” of apelin-13 in reperfusion of the heart [49]. The cardioprotective effect of apelin-12 could be mediated via the activation of antioxidant defense enzymes. Similar data were obtained by Azizi et al. [31]. They found that the post-infarction administration of Pyr^1^-apelin-13 (10 nmol/kg) reduced serum MDA, nitrite, and nitrate levels [31]. Perfusion of the isolated rat heart with a solution of apelin-13 improved the recovery of contractile function of the myocardium [49]. Pretreatment with chelerythrine, LY294002, the PKCε inhibitor εV1-2, and the mitochondrial K_ATP_ channel (mitoK_ATP_-channel) blocker 5-hydroxydecanoate (5-HD) abolished an inotropic effect of apelin [49]. It should be noted there is some opinion that apelins can stimulate NDPH oxidase and increase superoxide production [86].

Apelin-13 (0.1 mg/kg) reduced the infarct size in rats with CAO (30 min) and reperfusion (2 h) by about 36% [87]. Pretreatment with the PI3-kinase inhibitor LY294002, the ERK1/2 inhibitor PD98059, and mitochondrial permeability transition pore (MPT) opener atractyloside completely abolished an infarct-reducing effect of apelin-13. Apelin-13 increased p-Akt, p-ERK1/2, and p-glycogen synthase kinase-3β (p-GSK-3β) levels in myocardial tissue. Investigators concluded that apelin-13 triggered the cardioprotective pathway including PI3-kinase, Akt, ERK1/2, GSK-3β, and MPT pore closing [87]. Similar data were obtained by Pisarenko et al. [88]. They found that the PLC inhibitor U-73122, chelerythrine, the Na^+^/H^+^ exchange inhibitor amiloride, and the Na^+^/Ca^2+^ exchange inhibitor KB-R7943 eliminated a positive inotropic effect of the apelin-12 analogue (140 µmol/L) in the reperfusion of the isolated rat heart. The ERK1/2 inhibitor UO126, the PI3 kinase blocker LY294002, the NOS inhibitor L-NAME, and 5-hydroxydecanoate abolished the positive inotropic effect of the apelin-12 analogue in the reperfusion of the heart. The same inhibitors eliminated an infarct-sparing effect of this peptide (0.35 µmol/kg) in vivo [88]. Consequently, PLC, PKC, the Na^+^/H^+^ exchanger, the Na^+^/Ca^2+^ exchanger, ERK1/2, PI3-kinase, NOS, and the mitoK_ATP_-channel are involved in the cardioprotective effect of apelin. The isolated perfused rat heart was subjected to global ischemia (30 min) and reperfusion (2 h) [53]. Apelin-13 (0.5 µmol/L) was infused during the first 20 min of reperfusion. Apelin-13 reduced infarct size by about 50% and improved the recovery of contractile function of the heart in reperfusion [53]. It was found that the matrix metalloproteinase (MMP) inhibitor GM6001, the PI3-kinase inhibitor LY294002, the epidermal growth factor receptor (EGFR) tyrosine kinase inhibitor AG1478, the Src kinase inhibitor PP2, the mitoK_ATP_ channel blocker 5-HD, and the soluble guanylyl cyclase (GC) inhibitor ODQ abolished a cardioprotective effect of apelin-13 [53]. These data demonstrate that MMP, EGFR, Src kinase, the mitoK_ATP_ channel, and GC are involved in the cardioprotective effect of apelin-13 (Table 3).

The cardioprotective effect of apelins in I/R of the heart is accompanied by blockage of the MPT pore and the activation of PI3-kinase, Akt, ERK1/2, NOS, SOD, glutathione peroxidase, MMP, EGFR, Src kinase, the mitoK_ATP_ channel, GC, PLC, PKC, the Na^+^/H^+^ exchanger, and the Na^+^/Ca^2+^ exchanger (Figure 3). 

## 5. The Synthetic Analogues of Apelins

We have reported above that natural apelin receptor agonists have a short half-life. The in vivo half-life of apelin-13 in blood is 2.3 min in rats [89]. The half-life of serum apelin-13 was 30 min after intraperitoneal administration [54]. The half-life of Pyr^1^-apelin-13 is about 24 min in rat plasma ex vivo [90]. Therefore, the long-term cardioprotective effect of a chronic daily single administration of elabela-32, apelin-13, and Pyr1-apelin-13 in animals with MI or pressure overload is surprising [55,58,60,62,66]. The duration of the hypotensive effect of apelin-13 and apelin-17 analogues to remain stable during enzymatic hydrolysis before reducing BP in mice was about 60 min [33]. It is quite obvious that apelins that remain stable during enzymatic hydrolysis with a long half-life have the greatest prospects for clinical use in the treatment of stroke, acute myocardial infarction, and in the prevention of adverse myocardial remodeling.

The antidiabetic effects of stable long-acting fatty acid modified apelin analogues, namely, pGlu(Lys8GluPAL)apelin-13 amide (Lys8GluPAL) and apelin-13 amide, have been studied. The administration of both peptides for 28 days reduced glucose levels, food intake, and body weight and increased plasma insulin concentrations [91]. Tran et al. synthesized apelin-13 analogues with an in vivo half-life of 3.7 h [92]. Further reports have been conducted on synthesizing apelin-13 analogues with a plasma half-life of 3–8 h [33].

## 6. Conclusions

Apelins can protect the heart against ischemic and reperfusion injury both in vivo and in vitro. This effect is associated with the inhibition of apoptosis and ferroptosis and stimulation of autophagy. The role of necroptosis and pyroptosis in the cardioprotective effects of apelins remain unclear. Apelins protect the heart against both ischemic and reperfusion injury. The cardioprotective effect of apelins could be used in clinical practice. It can be hypothesized that apelin could prevent ischemic cardiac injury in the cardioplegic arrest of the heart and mitigate reperfusion cardiac injury in patients with AMI and PCI. Apelins can protect the heart against adverse remodeling in animals with myocardial infarction and pressure overload. Worth studying is a decrease in the protective effect of apelins or the disappearance of their effect as their concentration is increased [43,46,51,57]. It is unclear whether this is a result of their interaction with other receptors (non-apelin receptors) or a result of the presence of peptide impurities. For many therapeutic molecules, it has been observed that their activity is only visible using an intermediated dose—i.e., at low or high doses [93]. The cardioprotective effect of apelins in I/R of the heart is accompanied by the inhibition of the MPT pore, GSK-3β, and activation of PI3-kinase, Akt, ERK1/2, NOS, SOD, glutathione peroxidase, MMP, EGFR, Src kinase, the mitoK_ATP_ channel, GC, PLC, PKC, the Na^+^/H^+^ exchanger, and the Na^+^/Ca^2+^ exchanger. 

## Figures and Tables

**Figure 1 pharmaceutics-15-01029-f001:**
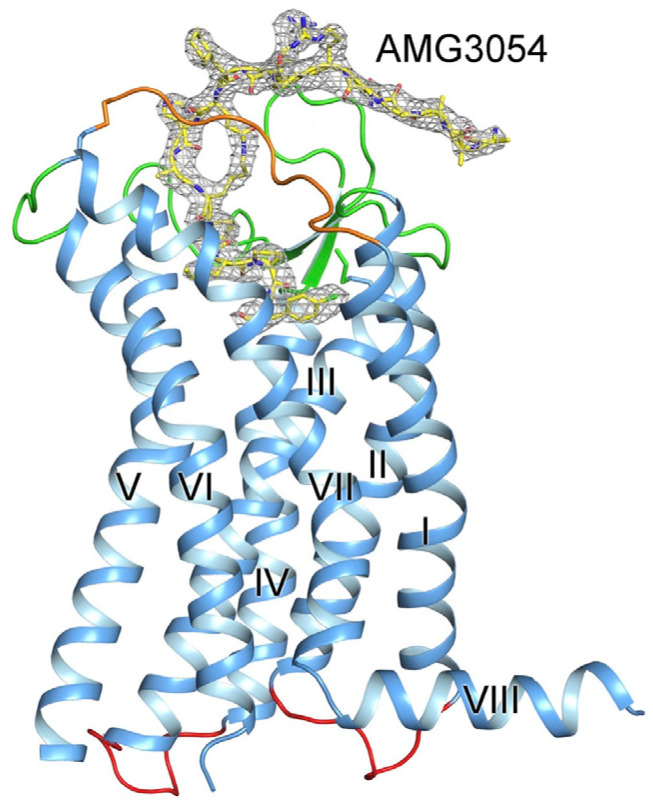
Structural determination of the apelin receptor with the peptide agonist AMG3054.

**Figure 2 pharmaceutics-15-01029-f002:**
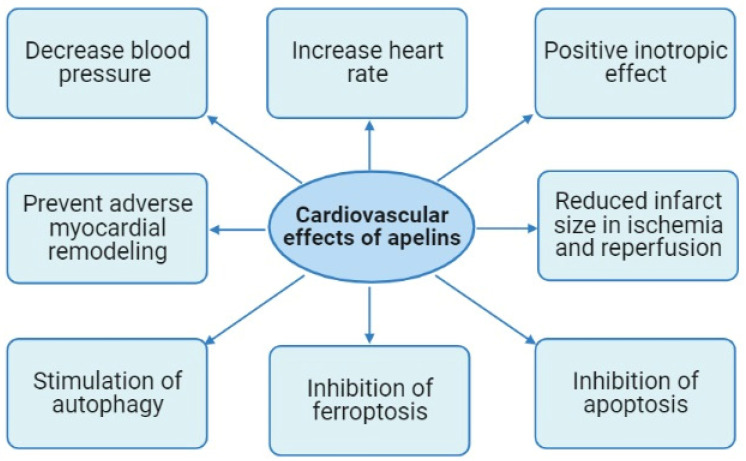
Cardiovascular effects of apelins.

**Figure 3 pharmaceutics-15-01029-f003:**
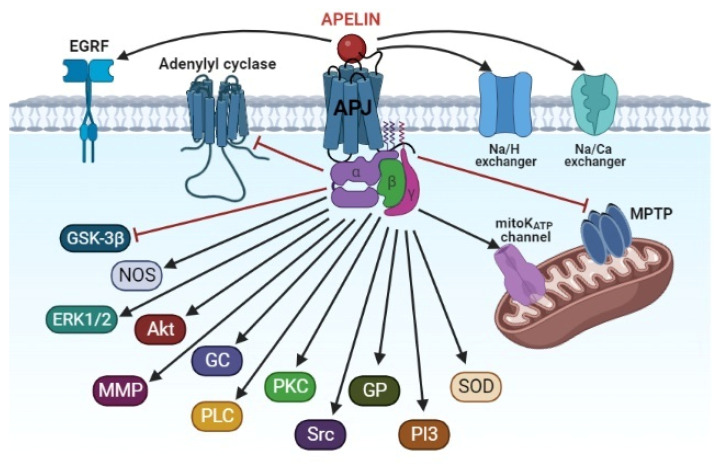
The signaling mechanism of the cardioprotective effect of apelins. Akt, protein kinase B; APJ, apelin receptor; EGRF, epidermal growth factor receptor; ERK1/2, extracellular regulated kinase; GC, guanylyl cyclase; GP, glutathione peroxidase; GSK-3β, glycogen synthase kinase-3β; MMP, matrix metalloproteinase; MPTP, mitochondrial permeability transition pore; NOS, NO-synthase; PI3, phosphoinositide-3-kinase; PKC, protein kinase C; PLC, phospholipase C; SOD, superoxide dismutase; Src, protooncogene, non-receptor tyrosine kinase.

**Table 1 pharmaceutics-15-01029-t001:** Amino acid sequences of apelin peptides.

Peptide	Amino Acid Sequences
Preproapelin	MNLRLCVQALLLLWLSLTAVCGGSLMPLPDGNGLEDGNVRHLVQPRGSRNGPGPWQGGRRKFRRQRPRLSHKGPMPF
Apelin-55	GSLMPLPDGNGLEDGNVRHLVQPRGSRNGPGPWQGGRRKFRRQRPRLSHKGPMPF
Apelin-36	LVQPRGSRNGPGPWQGGRRKFRRQRPRLSHKGPMPF
Apelin-17	KFRRQRPRLSHKGPMPF
Apelin-16	FRRQRPRLSHKGPMPF
Apelin-13	QRPRLSHKGPMPF
Pyr-apelin-13	Pyr-RPRLSHKGPMPF
Apelin-12	RPRLSHKGPMPF

Amino acid notation of apelin peptide isoforms is denoted using standard single letter amino acid codes. Pyr is pyroglutamate.

**Table 2 pharmaceutics-15-01029-t002:** The cardioprotective effect of apelins.

Apelins	Model	Doses	Administration	Effect	References
Apelin-13	Isolated mouse heart	1 µmol	During reperfusion	Reduced infarct size	[22,40,45,49]
		30 pmol/L	During reperfusion	Improvement of contractile function	
		30 pmol/L	During reperfusion	Reduced LDH and MDA	
Apelin-13	In vivo	1 mg/kg	In reperfusion	Reduced infarct size	[40,46]
		1 µg/kg	15 min before reperfusion	Prevention of necrosis and apoptosis	
Apelin-12	Isolated rat heart	140 μmol/L	Prior to ischemia or at the onset of reperfusion	Improvement of contractile function after ischemia Increase in ATP Reduced lactate and LDH	[43,47]
Apelin-12	In vivo	0.5 mg/kg	Before reperfusion	Reduced infarct size	[34,47]
Pyr-apelin-13	Isolated rat heart	10 nmol/L	Prior to ischemia or after ischemia	Reduced infarct size after ischemia No effect before ischemia No improvement of contractile function	[41]

ATP, adenosine triphosphate; LDH, lactate dehydrogenase; MDA, malondialdehyde.

**Table 3 pharmaceutics-15-01029-t003:** The signaling mechanism of the cardioprotective effect of apelins.

Apelins	Model	Doses	Mechanisms	References
Apelin-13 Apelin-36	COS cells	1 μmol	Stimulate Gi_1_ and Gi_2_ proteins, inhibit adenylyl cyclase, and cause the phosphorylation of ERK1/2 or Akt-kinase	[77]
Apelin-13	HEK293 cells	10 μmol	Activates Gq proteins and ERK1/2	[78]
	Isolated mouse heart	1 μmol	Activates PI3-kinase, Akt, and ERK1/2-kinases	[40,83]
	In vivo	1 μg/kg	Activates PI3 and ERK1/2-kinases, stimulates NOS	[46]
	Isolated rat heart	1 μmol	Activates PKCε, PI3-kinases, and the mitoK_ATP_-channel	[49]
	In vivo	0.1 mg/kg	Activates PI3-kinase, Akt, ERK1/2, GSK-3β, and MPT pore closing	[87]
	Isolated rat heart	0.5 µmol/L	Activates MMP, EGFR, Src kinase, the mitoK_ATP_ channel, and GC	[53]
Apelin-12	In vivo and in vitro	140 μmol/L	Stimulates NOS Reduces the MDA level in myocardial tissue, increases SOD, GP, catalase, and DMPO Activates PLC, PKC, the Na^+^/H^+^ exchanger, the Na^+^/Ca^2+^ exchanger, ERK1/2, PI3-kinase, NOS, and the mitoK_ATP_-channel	[43,44,47,85,88]
Pyr-apelin-13	HEK293 cells	80 nmol	Stimulates G_i/o_ and G_12/13_ proteins	[22]
	In vivo	10 nmol/kg	Reduces serum MDA, nitrite, and nitrate levels	[31]

Akt, protein kinase B; DMPO, 5,5-dimethyl-1-pyrroline-N-oxide; EGFR, epidermal growth factor receptor; ERK1/2, extracellular regulated kinase; GC, guanylyl cyclase; GP, glutathione peroxidase; GSK-3β, glycogen synthase kinase-3β; MDA, malondialdehyde; MMP, matrix metalloproteinase; MPTP, mitochondrial permeability transition pore; NOS, NO-synthase; PI3, phosphoinositide-3-kinase; PKC, protein kinase C; PLC, phospholipase C; SOD, superoxide dismutase; Src, protooncogene, non-receptor tyrosine kinase.

## Data Availability

Not applicable.

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
