# Peer review of "Apelin Is a Prototype of Novel Drugs for the Treatment of Acute Myocardial Infarction and Adverse Myocardial Remodeling"

_pharmaceutics, 2023, doi:10.3390/pharmaceutics15031029_

Round 1

Reviewer 1 Report

see attached file

Author Response

Dear colleague. Thank you very much for your comment. These comment helped us to improve quality of text. 

The manuscript of the review " Apelin is a Prototype of Novel Drugs for Treatment
of Acute Myocardial Infarction and Adverse Myocardial Remodeling.” by Popov.
et al. gives an overview of the use of Apelin as new potent therapeutic for acute
myocardial infarction (AMI) and remodeling.
The manuscript is in general well written and well structured. However, I have
some minor and major suggestions to improve this review for the scientific
committee.

  1. The authors present Apelin as a potent pharmacological treatment for AMI. However, during the whole manuscript, they present different models (with or without reperfusion) and administration protocols (before or after
    ischemia). In a view of a clinical application, only models with a reperfusion
    period are relevant with an administration after the ischemia (during or at
    the onset of arteria re-opening). This should be highlighted in the
    introduction or discussed at the end of the review.

We added this information in Introduction.

  1. The introduction should be in general longer – including more details on the epidemiology of AMI and a short description of other therapeutics having reach clinical studies. Also, an explanation concerning the differences between ischemic injury and reperfusion injury should be given.

We added this information in Introduction.

  1. Chapter 2 – could the authors add a table summarizing the sequences of the different Apelin peptides described in the review?

We added this table.

  1. Figure 1 – could the authors add to each effect the corresponding chapter of the review?

We corrected this figure.

  1. Information about the structural determination of Apelin receptor with a 17-mer mimetic peptide agonist is missing (DOI:htps://doi.org/10.1016/j.str.2017.04.008). A fgure with this structure
    could be added.

We added this figure.

  1. The description of the clinical study using Apelin to treat heart failure is
    missing (DOI: 10.1161/CIRCULATIONAHA.109.911339) – this could
    highlight the use of therapeutics targeting the Apelin receptor.

We added this information.

  1. Chapter 3.4 – could the authors add a table summarizing the conditions and the results for a better overview?

We could not prepare this table because we could not understand what you call the conditions and the results. What results and the conditions are you talking about?

  1. Chapter 4 - could the authors add a table summarizing the cardioprotective effects of Apelin for a better overview?

We added this table.

  1. In general, the authors should try to add some illustrations to their review.

We prepared three figures. Should we prepare the fourth Figure?

If so, what information should it contain?

  1. Conclusions – the authors note “Worth studying is a decrease in the
    protective effect of apelins or the disappearance of effect as their
    concentration is increased [39, 42, 47, 53]. It is unclear whether it is a result
    of their interaction with other receptor (non-apelin receptor) or a result of
    the presence of peptide impurities.”
    It is an observed phenomenon for many therapeutic molecules that its
    activity is only visible using an intermediated dose - at low or high doses
    the activity is lost. This is the U-shape curve of pharmacological drugs (see
    DOI: 10.1080/15287399009531412).
    Often the lost of activity at higher concentration is due to off-target effects or toxicity. I do not assume that it is due to peptide impurities (even if this
    could not be excluded).

We corrected conclusion and added Davis’s article.

Please add some comments on it in the conclusion section.

We added conclusion to each section.

Sincerely yours, Leonid N. Maslov

Reviewer 2 Report

The manuscript entitled “Apelin is a Prototype of Novel Drugs for Treatment of Acute Myocardial Infarction and Adverse Myocardial Remodeling”. The article’s topic sounds interesting and many articles related to apelin are very complete. But there are several parts from the manuscript that the authors need to revise. Below are some comments for the authors to check.

1.      Figure 1. Cardiovascular effects of apelins. The color and text of the picture are mostly blue, and it is suggested that the color should be changed.

2.      Page 4, 3.4 section. CAO must show the full name when they first time use.

3.      Page 5, line 16, I/R must show the full name when they first time use. Line 26 “increased TNF-α, IL-1β, IL-6 content “ change to “increased TNF-α, IL-1β, and  IL-6 content “.

4.      Page 9, mitoKATP channel in the Figure 2. ATP changed to subscript.

5.      Page 10, “The half-life of Pyr1-apelin-13 is about 24 min in rat plasma ex vivo [86].” change to font size 12. In the third line of the second paragraph, delete [87]. It is redundant.

6.      Delete  *** in the section 6 Conclusion.

Author Response

Dear colleague. Thank you very much for reviewing our article and important recommendations.

The manuscript entitled “Apelin is a Prototype of Novel Drugs for Treatment of Acute Myocardial Infarction and Adverse Myocardial Remodeling”. The article’s topic sounds interesting and many articles related to apelin are very complete. But there are several parts from the manuscript that the authors need to revise. Below are some comments for the authors to check.

  1. Figure 1. Cardiovascular effects of apelins. The color and text of the picture are mostly blue, and it is suggested that the color should be changed.

We corrected figure 1.

  1. Page 4, 3.4 section. CAO must show the full name when they first time use.

We corrected this error.

  1. Page 5, line 16, I/R must show the full name when they first time use. Line 26 “increased TNF-α, IL-1β, IL-6 content “ change to “increased TNF-α, IL-1β, and IL-6 content “.

We corrected this error.

  1. Page 9, mitoKATP channel in the Figure 2. ATP changed to subscript.

We corrected Figure 2.

  1. Page 10, “The half-life of Pyr1-apelin-13 is about 24 min in rat plasma ex vivo [86].” change to font size 12. In the third line of the second paragraph, delete [87]. It is redundant.

Font size was 12. We deleted [87].

  1. Delete *** in the section 6 Conclusion.

We corrected this error.

We cited new three references and changed text at the request of the first reviewer.

Sincerely yours, Leonid N Maslov

Round 2

Reviewer 1 Report

see attached file

Author Response

Dear colleague. Thank you very much for reviewing our article and important recommendations.

The review manuscript "Apelin is a Prototype of Novel Drugs for Treatment of Acute Myocardial Infarction and Adverse Myocardial Remodeling" by Popov. et al. provides an overview of the use of Apelin as a potent new therapy for acute myocardial infarction (AMI) and remodeling.

Based on the first round of review, the authors have added the missing information. However, I propose to modify two minor points

  1. Chapter 3.4 - could the authors add a table summarizing the conditions and results for a better overview?

The authors have added table 2 summarizing the results but the conditions are missing (doses and timing of administration). Could the authors add these to Table 2, please?

We added this table.

  1. Chapter 4 - could the authors add a table summarizing the cardioprotective effects of Apelin for a better overview?

The authors could add a table 3 summarizing the peptide used, the model used [cell culture (cell type?), ex vivo or in vivo (animal?)], the doses applied, the method of detection, the result obtained and the reference. A table similar to Table 2 but highlighting the signaling mechanism of Apelin function. Could the authors add a table like this?

We added this table.

Sincerely yours, Leonid N Maslov
